# Effectiveness of Benralizumab in OCS-Dependent Severe Asthma: The Impact of 2 Years of Therapy in a Real-Life Setting

**DOI:** 10.3390/jcm12030985

**Published:** 2023-01-27

**Authors:** Carolina Vitale, Angelantonio Maglio, Corrado Pelaia, Maria D’Amato, Luigi Ciampo, Giulia Pelaia, Antonio Molino, Alessandro Vatrella

**Affiliations:** 1Department of Medicine, Surgery and Dentistry “Scuola Medica Salernitana”, University of Salerno, 84100 Salerno, Italy; 2Department of Health Sciences, University “Magna Græcia” of Catanzaro, 88100 Catanzaro, Italy; 3Department of Respiratory Medicine, Federico II University, 80100 Naples, Italy

**Keywords:** severe asthma, OCS-dependent asthma, benralizumab, airway inflammation, anti-IL 5, corticosteroid resistance

## Abstract

Patients with severe OCS-dependent asthma can be considered a subgroup of asthma patients with severe disease and great risk of complications, related to chronic OCS use. The introduction of biological drugs has represented a turning point in the therapeutic strategy for severe asthma, offering a valid alternative to OCS. Benralizumab, like other anti-IL-5 agents, has been shown to reduce exacerbations and OCS intake/dosage and improve symptom control and lung function. While these findings have also been confirmed in real-life studies, data on long-term efficacy are still limited. Methods: In this retrospective study, we evaluated the effects of 2 years of treatment with benralizumab on 44 patients with OCS-dependent severe asthma by analyzing clinical, biological and functional data. Results: After 2 years of benralizumab, 59.4% discontinued OCS and patients who continued to use OCS had their mean dose reduced by approximately 85% from baseline. Meanwhile, 85% of patients had their asthma well-controlled (ACT score > 20) and had no exacerbations, and 41.6% had normal lung function. Conclusions: Our findings support the long-term effectiveness of benralizumab in severe OCS-dependent asthma in a real-life setting, suggesting potential reductive effects on costs and complications such as adverse pharmacological events.

## 1. Introduction

Severe asthma is a heterogeneous condition characterized by poor symptom control, recurrent exacerbations and frequent use of systemic oral corticosteroids (OCS), despite maximum optimized inhaled therapy [1,2]. OCS are prescribed as short-term therapy in exacerbations but also as maintenance therapy for symptom control in a significant number of patients with poorly controlled asthma. The proportion of severe asthma patients receiving OCS maintenance was reported to range from 20% to 60% [3]. Patients with asthma who need daily administration of OCS are termed OCS-dependent. These patients have a greater disease burden compared to those with OCS-independent asthma [4]. The consequences of long-term OCS therapy include the risk of developing different complications such as diabetes, infections, osteoporosis, cardiovascular diseases, gastrointestinal disorders and psychiatric disorders. Several studies have demonstrated that the risk of side effects is associated with the daily dose [5]. In addition, chronic use of OCS was significantly associated with all-cause mortality and also with an increase in healthcare resources’ utilization and costs [6,7]. It is clear that clinicians should pay attention to the benefit–risk ratio before considering long-term OCS treatment. However, despite the well-known and significant side effects, OCS continue to be a key therapeutic option for asthma treatment in clinical practice, due to their anti-inflammatory effects. As is known, the primary action of corticosteroids is to turn off multiple inflammatory genes, including those of cytokines, adhesion molecules and inflammatory mediator receptors. Corticosteroids are particularly effective on the T2 inflammatory pathway, where they target several elements, such as eosinophils, IL-4 and IL-5, leading to a rapid reduction in eosinophilic inflammation. Nevertheless, in some patients with severe asthma, persistent T2 inflammation occurs despite regular use of OCS. This condition is a frequently observed feature in patients affected by severe asthma. Steroid-resistant asthma is defined by a failure to improve the forced expiratory volume in 1 s (FEV1) by >15% even after a high dose of prednisolone for 2 weeks.

The inflammatory substrate observed in patients with severe asthma is heterogeneous. Alongside the well-described forms of T2-type inflammation, there are variants with a predominantly non-T2 inflammatory phenotype, in which neutrophilic inflammation is present (mainly driven by Th17 lymphocytes), or in which a predominant inflammatory substrate is not evident (paucigranulocytic) [8].

For the latter phenotypes, there are no active drug treatments currently approved for use in clinical practice, although promising randomized clinical trials are ongoing. Conversely, in recent years, a better understanding of the cellular mechanisms underlying T2 airway inflammation in asthma has led to the development of biological drugs that target elements of this specific inflammatory pathway [8,9,10]. Molecular targets of available biologics are: immunoglobulin E (IgE), interleukin-5 (IL-5) or interleukin-5 receptor (IL-5R), interleukin 4 (IL-4), interleukin 13 (IL-13) and thymic stromal lymphopoietin (TSLP). Among them, IL-5 plays the key pathogenetic role in T2 inflammation, being responsible for relevant pro-inflammatory effects including eosinophil recruitment and activation [11,12].

This cytokine has been identified as a strategic target of three biological drugs currently available for severe asthma treatment: mepolizumab, reslizumab and benralizumab 13]. Mepolizumab and reslizumab target IL-5 while benralizumab targets the IL-5 receptor alpha (IL-5Rα) subunit [12,13].

In OCS-dependent severe asthma patients, the introduction of anti-IL-5 biological drugs has offered a new therapeutic strategy that allows for a reduction in or suspension of OCS.

Benralizumab is an IgG1/k class humanized monoclonal antibody targeting the IL-5 receptor, indicated for add-on maintenance treatment of patients with severe eosinophilic asthma [14]. The efficacy of this anti-IL5 was demonstrated in large randomized controlled clinical trials [15,16,17]. These studies documented that the antieosinophilic effect was associated with a significant reduction in OCS intake/dosage, reduced exacerbations and improved asthma control and lung function, with a good safety and tolerability profile [15,16,17,18]. 

Data on the efficacy of benralizumab on patients with OCS-dependent severe asthma were provided by the ZONDA and PONENTE studies [17,19]. In these studies, benralizumab was shown to be effective in controlling asthma symptoms during OCS tapering and also after OCS discontinuation for 28 weeks in ZONDA and 36 weeks in PONENTE [17,19]. Specifically, in the ZONDA study, the median reduction in OCS dose was 75% for patients treated with benralizumab compared to 25% for the placebo. Over half (52%) of benralizumab-treated patients were able to discontinue the use of OCS completely, compared with 19% who received the placebo. In addition, a 70% reduction in the annual exacerbation rate was observed, compared to none for the placebo. These findings were strengthened by those of the PONENTE study. In that study, benralizumab eliminated maintenance OCS use in 62% of patients with OCS-dependent severe asthma. The majority (81%) of patients achieved complete elimination or were able to reduce the daily dose of OCS to 5 mg or less when further reduction was not possible due to adrenal insufficiency. Additionally, 74% of patients were free of asthma exacerbations during the tapering phase. The beneficial effects of benralizumab have recently been confirmed in real-life studies [20,21,22,23,24], although long-term effectiveness data are still limited. Here, we report the results obtained with benralizumab in OCS-dependent severe asthma patients treated for 2 years in a real-life setting.

## 2. Materials and Methods

This retrospective study included patients with OCS-dependent severe eosinophilic asthma, treated with benralizumab. Patients were enrolled consecutively during regular visits between December 2019 and April 2022 at 3 sites in Southern Italy. The diagnosis of severe asthma was made according to the European Respiratory Society (ERS)/American Thoracic Society (ATS) guidelines, and eligibility for benralizumab treatment was assessed according to the Italian Drug Agency (AIFA)’s prescription criteria. In addition to eligibility for benralizumab, other criteria for study participation were: OCS dependency and the exclusion of concomitant respiratory diseases and other conditions requiring maintenance steroid therapy. In our study, OCS dependency was defined as OCS maintenance treatment lasting 3 months or more in the previous year. Benralizumab was administered subcutaneously at a dosage of 30 mg every 4 weeks for the first three doses and every 8 weeks thereafter. After 4 weeks of first benralizumab administration, patients were instructed to gradually decrease OCS (by prednisone-equivalent 5 or 2.5 mg per week depending on baseline OCS dosage), until either discontinuation or the lowest dose required to maintain asthma control. All data were recorded on a common database. Follow-up assessment occurred at 6, 12, 18 and 24 months of treatment with benralizumab and included clinical and biological data collection and pulmonary function tests. The ACT score, number of exacerbations and need for systemic corticosteroid maintenance were recorded as clinical outcomes. Exacerbations considered were those with worsening of symptoms, requiring an increase in oral corticosteroids from a stable maintenance dose, for at least three days. The blood eosinophil count was recorded as the biological outcome and FEV1%, FVC%, FEV1/FVC% and FEF25-75% were reported as parameters of the pulmonary function. 

All the patients gave informed consent for the use of their personal data. This observational study was undertaken in accordance with the Declaration of Helsinki and was approved by the local ethical committee. 

### Statistical Analysis 

The statistical analysis was performed using Prism Version 9 (GraphPad Software Inc., San Diego, CA, USA). The data were reported as the mean and standard deviation (SD) for normally distributed data and median and interquartile range (IQR) for skewed distributed data. The categorical variables were considered as the number of cases and percentages. The Anderson–Darling test was conducted to investigate whether data were normally distributed. Dunnett’s multiple comparison test and Friedman’s test were used to compare variables, when appropriate. Statistical significance was set at a threshold of *p* value < 0.05.

## 3. Results

Our retrospective observational study analyzed 44 patients with OCS-dependent severe asthma treated with benralizumab. Two years after initiation of treatment, data from 32 patients were available for evaluation. Some clinical and biological information was not obtainable for all patients at the time of collection due to the retrospective nature of the study. Baseline characteristics of our patients are reported in Table 1. All the patients had been treated continuously with OCS for 3 months or more before enrollment. Our study population was predominantly female (27, 63.8%), with a mean age of 59.5 years and a mean BMI of 27.1 kg/cm^2^. Most patients had a history of smoking (59.1%), 54.5% of the patients were defined as allergic based on skin prick test (SPT) positivity and the most reported comorbidities were gastroesophageal reflux disease (43%) and chronic rhinosinusitis with nasal polyposis (41%). All patients were receiving high-dose inhaled corticosteroids (ICS) and a long-acting β2-agonist (LABA) combination. About half of the patients (n 23, 52.27%) also required add-on bronchodilator therapy with long-acting muscarinic antagonists (LAMA), and about one-third (13, 29.5%) of the patients used leukotriene receptor antagonists. All patients were receiving OCS maintenance treatment with a median dose of 12.5 mg/day prednisone-equivalent (IQR:20). All patients had a history of poorly controlled asthma, as suggested by an ACT score < 20 (mean value 12.9), with frequent exacerbations despite maximum therapy (mean 5.5 in the previous year). In addition, all patients had airflow limitation, most often of moderate degree (mean predicted FEV1 57.18%). Regarding blood eosinophils, at baseline, the median eosinophil count was 695 (IQR: 508). Other causes of hypereosinophilia were excluded in 3 patients with blood eosinophilic counts > 1500 cells/μL, prior to initiating benralizumab treatment. 

### 3.1. Blood Eosinophil Counts

Blood eosinophil counts dropped to zero after benralizumab treatment. These data were recorded 6 months after the start of treatment and for the entire duration of the study.

### 3.2. Lung Function

Treatment with benralizumab resulted in a significant improvement in all lung function parameters. At baseline, all patients had airflow limitation (FEV1/FVC 59.7% ± 9.2), with a mean FEV1 of 57.2%, and small airway involvement, with mean a FEF 25–75% of 35.2%. The mean percentage increases in FEV1/FVC and FEV1 recorded 6 months after the initiation of treatment were 7.8% and 18.42%, respectively, and FEV1 was >80% in 40.6% of patients. This improvement was also recorded for small airways with a mean percent increase in FEF 25–75% of 17.6%. That trend was maintained throughout the study period, and after 2 years of treatment, 41.6% of patients had normal lung function (Table 2, Figure 1).

### 3.3. Exacerbations

Benralizumab significantly reduced asthma exacerbations. At the start of treatment, the annual exacerbation rate was 5.54. After 6 months of treatment, 87.2% of patients had no exacerbations. This result was largely maintained after 24 months with 85% of patients free of exacerbations. The annual exacerbation rate dropped from 5.54 to 0.18 and 0.23 after 1 and 2 years of treatment, respectively (Table 2, Figure 2). 

### 3.4. Asthma Control

Benralizumab improved asthma control. After 6 months of therapy, asthma was well-controlled in 77% of patients (ACT ≥ 20), with a mean increase in ACT score of 8.4 points from baseline. This result was further improved after 24 months of treatment, with 85% of patients having an ACT score ≥ 20 and a mean increase of 8.9 points (Table 2, Figure 2).

### 3.5. OCS Use

Treatment with benralizumab led to discontinuation of OCS use in the majority of patients. After 6 months of treatment, users of OCS had dropped from 100% to 43.5%, and after 24 months, only 40.6% of patients continued to use them.

During treatment with benralizumab, for patients who continued OCS, the mean dose they used decreased by approximately 85% from baseline to 6 months and 24 months (Table 2, Figure 2).

### 3.6. Safety

Benralizumab was shown to be safe and well-tolerated. One patient reported injection site erythema, and besides that, no other side effects were reported.

## 4. Discussion

Our findings support the long-term effectiveness of benralizumab in OCS-dependent severe asthma in a real-life setting. After 2 years of benralizumab, 85% of patients had well-controlled asthma (ACT score > 20) and were free from exacerbations, 41.6% had normal lung function, 59.4% discontinued OCS and those patients who continued to use OCS had significantly reduced the drug dose taken. 

These effects are related to the ability of benralizumab to counter eosinophilic inflammation targeting eosinophils through a dual mechanism [14]. The main process is to bind the alpha chain of the IL-5 receptor, selectively expressed on eosinophils and basophils, inhibiting its binding to IL-5 and thus neutralizing the fundamental survival signal of these cells. The second is to induce apoptosis of eosinophils by directly activating antibody-dependent cell-mediated cytotoxicity (ADCC) through interaction with the FcγIIIRa receptor expressed by the NK cell [14]. Through these mechanisms, benralizumab can deplete blood eosinophils and induce a drastic decrease in airway eosinophils [14]. In our study, at baseline, all patients presented persistent high peripheral blood eosinophils despite maintenance therapy with OCS. The reasons for persistent eosinophilia in patients with asthma treated with OCS are unclear, but it is probably due to an impaired biological response to corticosteroids, which can result in steroid resistance or insensitivity. Steroid resistance is defined by a failure to improve the forced expiratory volume in 1 s (FEV1) by >15% even after a high dose of prednisolone for 2 weeks. Conversely, steroid insensitivity describes a condition in which patients need to take OCS to maintain asthma control [25]. Defects in the ability of the glucocorticoid receptor (GR) to bind to the drug and translocate to the nucleus, excessive activation of activating peptide-1 (AP-1) and NF-κB, impaired histone deacetylase and increased dysfunctional GRβ isoform expression are the main pathways involved in corticosteroid resistance/insensitivity [11,26]. 

Benralizumab completely depleted peripheral blood eosinophils, and this effect was sustained over time, even in patients who withdrew OCS. The biological action on eosinophils has extremely positive effects both from clinical and functional points of view.

Eosinophilic airway inflammation is closely related to the risk of exacerbation. In our study, benralizumab induced a marked reduction in exacerbations. The annualized exacerbation rate dropped from 5.54 to 0.18 and 0.23 after 1 and 2 years of treatment, respectively. After 2 years of benralizumab, 85% of patients were exacerbation-free. This finding confirms what was recently reported in a real-life study over a period of approximately 20 months [24] and supports the long-term positive effects of benralizumab, even in OCS-dependent asthma patients. 

Furthermore, benralizumab reduced airflow limitation, as demonstrated by the increases in FEV1, the FEV1/FVC ratio and FEF 25–75%. The improvement was observed after 6 months and remained stable over time for all 3 parameters. Unlike the results recently reported in another real-life study [24], improvement in small airway obstruction did not slow over time. The effects of benralizumab on small airway disease were assessed in 2 retrospective studies, which demonstrated significant improvements in FEF 25–75% after 24 weeks of treatment [27,28]. The positive effect of anti-IL-5 biologic drugs on peripheral airways was also supported by a recent study demonstrating the efficacy of mepolizumab in patients with eosinophilic severe asthma [29].

Regarding patient-reported asthma control, benralizumab induced a rapid improvement in symptoms that was sustained over time, as evidenced by the significant increase in the ACT score at all visit times. In 77% of cases, the ACT score reached the cut-off of 20 points after 12 weeks of treatment. This percentage increased to 85% after 2 years of treatment. These results confirm the previous findings on the effectiveness of benralizumab in symptom control [20,21,22,23,24].

The clinical and functional improvements described above allowed most patients to discontinue OCS. 

After 6 months of treatment, 56.9% of patients stopped OCS, and the remaining 43.1% reduced their dose by approximately 85% from baseline. These percentages were maintained at subsequent follow-ups, and at 24 months of treatment, 59.4% of patients did not take OCS, and the remaining 40.6% had an approximately 85% dose reduction from baseline.

In our study, the proportion of patients who withdrew OCS was higher than previously reported. Pelaia et al. observed about 50% discontinuation after 6 months of therapy [21]. A similar proportion was reported after 12 months of treatment with benralizumab by Kavanagh et al. in a real-world experiment with a large cohort of patients with severe eosinophilic asthma [20]. The authors observed that benralizumab led to significant reductions in asthma exacerbations despite a median reduction in OCS dose of 100%, with 70% of patients able to stop OCS therapy for asthma. In the ANANKE study, which examined a large sample of patients over a treatment period with a median duration of 9 months, OCS therapy was interrupted in 43.2% of cases and the dose reduced by 56% [23]. Recently, Sposato et al. reported OCS discontinuation for 48.4% of patients after a mean period of 20 months of benralizumab [24]. The reasons for the higher OCS discontinuation rate in our study are likely to lie in the baseline characteristics of our patients. Blood eosinophil count ≥300 cells/μL, presence of nasal polyposis, adult-onset asthma, use of OCS, pulmonary function impairment and a history of ≥3 exacerbations/year have recently been identified as baseline characteristics predictive of an enhanced response to benralizumab [30,31,32,33]. All these characteristics were found in all of our patients, with the exception of nasal polyposis, which was found in less than half of the cases (41%). 

This study had some limitations. Adrenal function was not assessed. As is known, long-term use of OCS can lead to adrenal insufficiency due to suppression of the hypothalamic–pituitary–adrenal axis. In the PONENTE study, adrenal insufficiency was detected in 60% of patients at the first evaluation and in 38% of patients 2–3 months later. However, most OCS-dependent patients with severe asthma treated with benralizumab were able to stop or maximally limit corticosteroids with careful management. In our study, patients had no symptoms indicative of adrenal insufficiency, and OCS treatment was stopped or reduced maximally after tapering. However, in clinical practice, adrenal function should be monitored regularly in patients who have their OCS interrupted or reduced while receiving benralizumab, even in patients with apparently normal function.

This is because the dynamic adrenocorticotropic hormone (ACTH) stimulation test, used to evaluate the degree of adrenal reserve impairment, may give apparently normal results in some cases and expose patients to the risk of adrenal reserve impairment in response to stressful stimuli [34,35]. In addition, the number of patients enrolled is relatively small and the absence of an active comparator arm or placebo arm makes drawing formal conclusions difficult. However, our findings may be of interest for several reasons. Our results provide evidence of the beneficial effects of long-term treatment with benralizumab in patients with OCS-dependent severe asthma, after OCS discontinuation or reduction, in a real-life setting. After 2 years of treatment, all of our patients can be considered therapy responders, and 56.3% of patients can be considered super responders, based on the absence of exacerbations, OCS discontinuation and significant improvements in asthma control, as recently defined by a Delphi panel [36]. These pieces of evidence support the data previously published in the literature and enrich their validity through the long period of observation. Furthermore, they concern patients with OCS-dependent severe asthma. These patients can be considered a subgroup of severe asthmatics with more severe disease and greater risk of complications, related to chronic use of OCS. According to the 2022 GINA document, the use of OCS for maintenance therapy should be considered as a last resort and only recommended in low doses and as short-term as possible in patients with severe, uncontrolled asthma despite maximum therapy, including biologics [1]. In addition, GINA recommends that patients with severe asthma taking OCS should be counseled about potential side effects resulting from chronic use, and gradually reduce their dose until therapy is stopped, when possible, in order to prevent the consequences of long-term OCS therapy [1]. Nevertheless, OCS are used more widely in clinical practice than recommended by guidelines due to their therapeutic effect and also because they are readily available, low-cost and do not require a specialized prescribing center, unlike biologics. 

In our study, the adjunctive therapy with benralizumab, while leading to a substantial reduction in and sometimes suspension of maintenance OCS, did not allow us to avoid its use in all cases. Indeed, the regular use of OCS in the treatment of patients with severe asthma raises several concerns regarding the proper management of therapy.

Although OCSs have been used for decades in severe asthma, the optimal duration or dose to control the disease has not been determined, nor are there standardized protocols for discontinuation. In addition, the occurrence of undesirable effects should be monitored, and in particular, adrenal function should be evaluated. 

All these elements should be taken into consideration by physicians when prescribing long-term OCS therapy. These considerations are particularly valid for patients with severe asthma since a proportion of these patients requires maintenance OCS despite adjunctive biological therapy.

The introduction of biologics has represented a turning point in the therapeutic strategy for severe asthma, offering a relevant alternative to OCS, which was until recently the mainstay of asthma treatment. Benralizumab, like other anti-IL-5 agents, has been shown to improve symptom control and lung function and reduce exacerbations, allowing OCS treatment to be discontinued or tapered and preventing the potential risk of adverse events of OCS maintenance therapy [20,21,22,23,24]. 

## 5. Conclusions

Substantial progress in understanding the pathogenic mechanisms underlying the heterogeneous endotypes of severe asthma has led to the development, approval and practical use of many biologics, the efficacy and safety of which have been convincingly demonstrated by both RCTs and, more recently, by real-life observations.

Adding to the literature, our real-life study demonstrates that long-term treatment with benralizumab is particularly and persistently effective in a population of OCS-dependent patients with severe eosinophilic asthma. Benralizumab demonstrated progressive and sustained efficacy on symptoms, exacerbations and pulmonary function, and these benefits were observed despite discontinuation of or a drastic reduction in OCS use. Therefore, alongside the consistent clinical-functional benefits resulting from the direct action of the drug on the inflammatory substrate, benralizumab, thanks to its OCS-sparing effect, also produces advantageous effects in terms of reducing possible systemic side effects deriving from chronic use of steroids. This translates into better management of severe asthma both from a clinical point of view and in terms of direct and indirect costs.

## Figures and Tables

**Figure 1 jcm-12-00985-f001:**
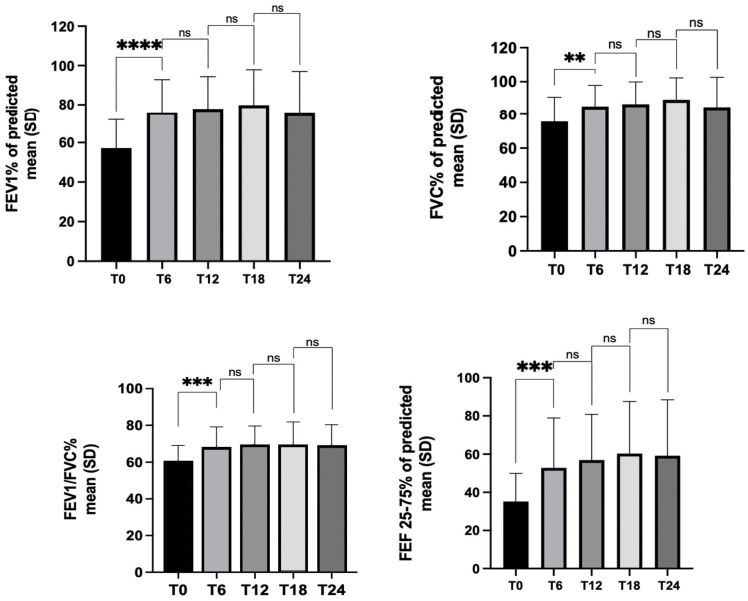
Effects of benralizumab on pulmonary function tests over time (at baseline (T0) and then at 6 (T6), 12 (T12), 18 (T18) and 24 (T24) months of follow-up). ** *p* = 0.048, *** *p* = 0.0005, **** *p* = 0.0001, ns = not statistically significant (Dunnett’s multiple comparison test).

**Figure 2 jcm-12-00985-f002:**
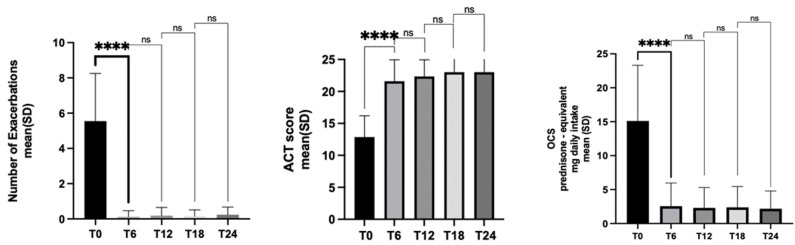
Effects of benralizumab on exacerbations, symptoms control and OCS intake over time (at baseline (T0) and then at 6 (T6), 12 (T12), 18 (T18) and 24 (T24) months of follow up). **** *p* = 0.0001, ns = not statistically significant (Dunnett’s multiple comparison test).

**Table 1 jcm-12-00985-t001:** Baseline characteristics of severe asthmatic patients.

Patients, n	44
Age (mean ± SD)	59.4 ± 9.5
Gender (female)	27 F (61.3%)
Smokers/former smokers/non-smokers	2/26/16
Body mass index (kg/m^2^, mean ± SD)	27.1 ± 7.32
Asthma duration median (IQR)	21.5 (16.5)
Age at asthma onset (mean ± SD)	36.3 ± 13.5
Atopy, n	24 (54.5%)
OCS-dependent patients	44 (100%)
** *Comorbidities* **	
Obesity (BMI ⩾ 30 kg/m^2^)	6 (13.6%)
Chronic rhinosinusitis with nasal polyposis	18 (41%)
Gastroesophageal reflux disease	19 (43.1%)

**Table 2 jcm-12-00985-t002:** Clinical, biological and functional data on patients *.

	T0	T6	T12	T18	T24
Patients, n	44	44	39	36	32
ACT score, mean ± SD	12.8 ± 3.3	21.6 ± 3.3	22.3 ± 2.6	23 ± 2.3	23 ± 2.2
Blood eosinophils (cells/μL), median (IQR)	695 (508)	0	0	0	0
Exacerbation history, previous year (n/y), mean ± SD	5.54 ± 2.7	0.12 ± 0.34	0.18 ± 0.46	0.15 ± 0.36	0.23 ± 0.44
OCS users, n	44 (100%)	19(43.1%)	17 (43.5%)	14 (38.8%)	13 (40.6%)
OCS (prednisone-equivalent mg/day), mean ± SD	15.2 ± 12.5	2.3 ± 3	2.3 ± 2.9	2.4 ± 3	2.2 ± 2.6
FEV1, %th mean ± SD	57.2 ± 14.5	75.6 ± 17.2	77.8 ± 27.6	79.8 ± 38.1	75 ± 37.8
FVC, %th mean ± SD	75.7 ± 15.1	85.3 ± 12.5	86.7 ± 28.2	89.4 ± 40.4	85 ± 40.1
FEV1/FVC, %th mean ± SD	59.7 ± 9.2	67.5 ± 11.6	68.6 ± 23.1	68.5 ± 32.8	68.5 ± 31.6
FEF 25–75%, %th mean ± SD	35.2 ± 14.7	52.8 ± 26	56.9 ± 28.1	60.3 ± 35.3	59.2 ± 29.2

* Dunnett’s multiple comparison test was used to compare the data: the results are reported in Figure 1 and Figure 2.

## Data Availability

Data are available on request due to privacy restrictions.

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
