# Peer review of "Effectiveness of Benralizumab in OCS-Dependent Severe Asthma: The Impact of 2 Years of Therapy in a Real-Life Setting"

_jcm, 2023, doi:10.3390/jcm12030985_

Round 1

Reviewer 1 Report

I would like to thank the authors for allowing me to review their work and commend them on a job well done. In this retrospective analysis, the authors present long term real life efficacy data of Benra on OCS dependent asthma. Their results are remarkable. Partly, as they have disclosed, due to the fact that they captured at baseline, a sick group of OCS dependent asthma patients with high T2 signals. However, these patients do exist and it important for such papers to continue to contribute their experiences to the larger. The study was well designed, the results clearly presented and the paper well written. 

Major comments

None

Minor Comments 

1.    Consider changing the term asthmatics to asthma patients or patients with asthma. 

2.    As a comparison, could the authors include OCS related outcomes from Zonda and Ponente (lines 69-70) ie. OCS free patients, OCS dose reduction etc.

3.    Was adrenal insufficiency considered during the analysis? If not, maybe the authors could mention it as a note in their discussions. As noted in the PONENTE study, AI can be noted in a relatively large number of OCS dependent asthma patients. Clinicians should be (continuously) cautioned about monitoring for AI in OCS sparing studies as a reminder.

4.    Minor typo on Line 102-103. “data. data” .

5.    Typo on line 124. It should be 12.5mg / day not mg/die

6.    Same typo noted in table 2 - mg / day not mg/die

7.    There are formatting issues with the references 

Author Response

We want to thank you for your interesting and valuable comments, which have allowed us to improve our manuscript.

We modified the manuscript following your suggestions.

Minor Comments 

  1. Consider changing the term asthmatics to asthma patients or patients with asthma.

       We changed the term asthmatics with asthma patients or patients with  asthma.

  1. As a comparison, could the authors include OCS related outcomes from Zonda and Ponente (lines 69-70) ie. OCS free patients, OCS dose reduction etc. 

We reported OCS related outcomes from Zonda and Ponente in the introduction:

"Specifically, in the ZONDA study, the median reduction in OCS dose was 75% for benralizumab-treated patients compared with 25% for placebo. 52% percent of benralizumab-treated patients were able to discontinue use of OCS completely, compared with 19% who received placebo. In addition, a 70% reduction in the annual exacerbation rate was observed compared to placebo. These findings were strengthened by those of the PONENTE study. In this study, benralizumab eliminated maintenance OCS use in 62% of patients with severe OCS-dependent and 81% of patients achieved complete elimination or were able to reduce the daily dose of OCS to 5 mg or less when further reduction was not possible due to adrenal insufficiency. Additionally, 74% of patients experienced no asthma exacerbations during the tapering phase".

  1. Was adrenal insufficiency considered during the analysis? If not, maybe the authors could mention it as a note in their discussions. As noted in the PONENTE study, AI can be noted in a relatively large number of OCS dependent asthma patients. Clinicians should be (continuously) cautioned about monitoring for AI in OCS sparing studies as a reminder. 

Unfortunately, adrenal function could not be assessed during the study. However, following your suggestion, we addressed this in the discussion, emphasizing the importance for clinicians to monitor adrenal function in patients with severe OCS-dependent asthma who are weaned off oral corticosteroids while taking benralizumab.

“Adrenal function could not be assessed. As is known, long-term use of OCS can lead to adrenal insufficiency, due to suppression of the hypothalamic-pituitary-adrenal axis. In the PONENTE study, adrenal insufficiency was detected in 60% of patients at the first evaluation and in 38% of patients 2-3 months later. However, most OCS-dependent patients with severe asthma treated with benralizumab were able to stop or maximally limit corticosteroids with careful management. In our study, patients had no symptoms indicative of adrenal insufficiency, and OCS treatment was stopped or tapered maximally after tapering. However, in clinical practice, adrenal function should be monitored regularly in patients who have their OCS interrupted or reduced while receiving benralizumab, even in patients with apparently normal function. This is because the dynamic adrenocorticotropic hormone (ACTH) stimulation test, used to evaluate the degree of adrenal reserve impairment, can give apparently normal results in some cases and expose patients to the risk of adrenal reserve impairment in response to stressful stimuli (references: Lipworth B, Misirovs R, Chan R. Adrenal insufficiency in patients taking benralizumab as corticosteroid sparing therapy. Lancet Respir Med. 2022 Jan;10(1):e7., Broide J, Soferman R, Kivity S, et al. Low-dose adrenocorticotropin test reveals impaired adrenal function in patients taking inhaled corticosteroids. J Clin Endocrinol Metab 1995; 80: 1243–46.). 

  1. Minor typo on Line 102-103. “data. data”. We  corrected it.

  1. Typo on line 124. It should be 12.5mg / day not mg/die. We replaced  “die” with day

  1. Same typo noted in table 2 - mg / day not mg/die. We replaced “die” with day

  1. There are formatting issues with the references. We checked and corrected the formatting errors of the references.

Reviewer 2 Report

The authors investigated effectiveness of benralizumab upon OCS-dependent adult ashtma patients in a real-life setting. They showed that benralizumab reduced OCS use, increased well-controlled asthma, reduced exacerbations and improved lung-function. The presented data are potentially relevant in the field of clinical practice of asthma. However, I have several concerns study description and analysis.

1. Please indicate the number of participants only in Results section but not in Materials and Methods.

2. Please indicate how the authors recruited the participants more precisely. What is eligibility of participants ? Are they consecutive cases in a single or multiple insitutions ? Are there anyl inclusion and exclusion criteria for the participants.  These are very importnat for the readers to understand the results of this clinical study. Please provide detailed information.

3. Statistical analysis. The authors compared multiple results in Figure 1, 2 and Table 2. But there is no descrittion of statistical analysis for multiple comparison, such as ANOVA and Kruskal-Wallis. Please do multiple comparition appropreately and indicate which methos are used in Figure legends and Table.

4. Figures. What do T0, T6,,,stands for. The authors should describe it in both figure legends.

Author Response

We want to thank you for your valuable comments, which have allowed us to improve our manuscript.

We modified the manuscript following your suggestions.

  1. Please indicate the number of participants only in Results section but not in Materials and Methods. We indicated the number of participants only in Results section.
  2. Please indicate how the authors recruited the participants more precisely. What is eligibility of participants ? Are they consecutive cases in a single or multiple insitutions? Are there anyl inclusion and exclusion criteria for the participants.  These are very important for the readers to understand the results of this clinical study. Please provide detailed information. We clarified how patients were recruited in the study and elegibility criteria in Materials and Methods Section.

“This retrospective study included patients with OCS dependent severe eosinophilic asthma, treated with benralizumab. Patients were enrolled consecutively during regular visit between December 2019 and April 2022 at 3 sites in Southern Italy. The diagnosis of severe asthma was made according to the European Respiratory Society (ERS)/American Thoracic Society (ATS) guidelines and eligibility for benralizumab treatment was assessed according to the Italian Drug Agency (AIFA)’s prescription criteria. In addition to eligibility for benralizumab, other criteria for study participation were: OCS dependency, exclusion of concomitant respiratory diseases and other conditions requiring maintenance steroid therapy.” 

  1. Statistical analysis. The authors compared multiple results in Figure 1, 2 and Table 2. But there is no descrittion of statistical analysis for multiple comparison, such as ANOVA and Kruskal-Wallis. Please do multiple comparition appropreately and indicate which methos are used in Figure legends and Table.

 We rewrote the section on statistical analysis to be clearer and more appropriate and indicated methods use for statistical analysis in Figure Legends and Table. 

“The statistical analysis was performed using Prism Version 9 (Graphpad Software Inc., San Diego, CA, USA). The data were reported as mean and standard deviation (SD) for normally distributed data and median and interquartile range (IQR) for skewed distributed data. The categorical variables were considered as the number of cases and percentages. Anderson-Darling test was applied to investigate if data were normally distributed. Dunnett’s multiple comparison test and Friedman test were used to compare variables, when appropriate. Statistical significance was set at a threshold of p value <0.05.”

4. What do T0, T6,,,stands for. The authors should describe it in both figure legends.

We clarified in both figure legend what T0, T6, T12, T18 and T24 stand for.

Figure 1. Effects of benralizumab on pulmonary function tests over time (at baseline (T0) and then at 6 (T6), 12 (T12), 18 (T18) and 24 (T24) months of follow up). (**, p= 0.048, ***, p= 0.0005, ****, p=0.0001, ns = not statistically significant,Dunnett’s multiple comparison test).

Figure 2. Effects of benralizumab on exacerbations, symptoms control and OCS intake over time (at baseline (T0) and then at 6 (T6), 12 (T12), 18 (T18) and 24 (T24) months of follow up). (****, p=0.0001, ns = not statistically significant,Dunnett’s multiple comparison test).

Round 2

Reviewer 2 Report

I confirmed the authors approapreately corrected the manuscript.